# Wildlife–Livestock Host Community Maintains Simultaneous Epidemiologic Cycles of *Mycoplasma conjunctivae* in a Mountain Ecosystem

**DOI:** 10.3390/vetsci11050217

**Published:** 2024-05-14

**Authors:** Jorge Ramón López-Olvera, Eva Ramírez, Carlos Martínez-Carrasco, José Enrique Granados

**Affiliations:** 1Servei d’Ecopatologia de Fauna Salvatge (SEFaS) and Wildlife Ecology & Health Group (WE&H), Departament de Medicina i Cirurgia Animals, Facultat de Veterinària, Universitat Autònoma de Barcelona (UAB), Bellaterra, E-08193 Barcelona, Spain; eva.ramirez.gr7@gmail.com; 2Departamento de Sanidad Animal, Facultad de Veterinaria, Universidad de Murcia, E-30100 Murcia, Spain; cmcpleit@um.es; 3Parque Nacional y Parque Natural de Sierra Nevada and Wildlife Ecology & Health Group (WE&H), Pinos Genil, E-18191 Granada, Spain; josee.granados@juntadeandalucia.es

**Keywords:** *Capra pyrenaica*, endemic, epidemiology, Iberian ibex, infectious keratoconjunctivitis, PCR, reservoir

## Abstract

**Simple Summary:**

Infectious keratoconjunctivitis is an eye disease caused by *Mycoplasma conjunctivae* that affects domestic and wild caprines, including Iberian ibex (*Capra pyrenaica*) and domestic sheep and goats. This study assessed *M. conjunctivae* in the host community of the Natural Space of Sierra Nevada (NSSN), a mountain habitat in southern Spain. *Mycoplasma conjunctivae* strains circulated endemically without causing clinical signs in Iberian ibex and livestock, either shared or maintained independently by each host species. In Iberian ibex, endemic infection was maintained by naïve subadults, with an epizootic outbreak when the infection spread to adults. Goat was at least as important as sheep in maintaining *M. conjunctivae*. The results suggest that the epidemiological role of wild ungulates should be considered in mountain ecosystems, as their mobility may contribute to the spread of IKC and other shared pathogens among spatially segregated livestock flocks.

**Abstract:**

Infectious keratoconjunctivitis (IKC) is an eye disease caused by *Mycoplasma conjunctivae* that affects domestic and wild caprines, including Iberian ibex (*Capra pyrenaica*), a medium-sized mountain ungulate. However, its role in IKC dynamics in multi-host communities has been poorly studied. This study assessed *M. conjunctivae* in Iberian ibex and seasonally sympatric domestic small ruminants in the Natural Space of Sierra Nevada (NSSN), a mountain habitat in southern Spain. From 2015 to 2017, eye swabs were collected from 147 ibexes (46 subadults, 101 adults) and 169 adult domestic small ruminants (101 sheep, 68 goats). *Mycoplasma conjunctivae* was investigated through real-time qPCR and statistically assessed according to species, sex, age category, year, period, and area. The lppS gene of *M. conjunctivae* was sequenced and phylogenetically analysed. *Mycoplasma conjunctivae* was endemic and asymptomatic in the host community of the NSSN. Three genetic clusters were shared by ibex and livestock, and one was identified only in sheep, although each host species could maintain the infection independently. Naïve subadults maintained endemic infection in Iberian ibex, with an epizootic outbreak in 2017 when the infection spread to adults. Wild ungulates are epidemiologically key in maintaining and spreading IKC and other shared diseases among spatially segregated livestock flocks.

## 1. Introduction

Infectious keratoconjunctivitis (IKC) is a highly contagious eye disease that affects domestic and wild caprines [1]. Diverse infectious agents, such as *Chlamydophila psittaci*, *C. abortus*, *C. pecorum*, *Moraxella ovis* (formerly *Branhamella ovis*), and different viruses, have been isolated from IKC-affected eyes [2,3,4,5,6,7,8,9,10,11]. However, *Mycoplasma conjunctivae* is currently accepted as the etiological agent of IKC [1,7,12,13,14,15,16,17,18,19,20,21], although other mycoplasmas might be involved, requiring molecular investigations for a correct diagnosis [9,22]. *Mycoplasma conjunctivae* can be transmitted through direct contact, air dispersion, and mechanical vectors, such as flies of the family Muscidae [23,24]. Clinical signs usually occur bilaterally and include ocular discharge, epiphora, mild conjunctivitis, and corneal opacity, causing transitory blindness in most cases. Brain lesions associated with infectious keratoconjunctivitis have also been described in wild mountain ungulates [25]. The animals affected by IKC usually recover spontaneously, but keratoconjunctivitis may also progress to staphyloma and perforation of the cornea when no treatment is applied, causing irreversible eye lesions and, consequently, permanent blindness [26]. To the authors’ knowledge, IKC in humans originating from zoonotic transmission of *M. conjunctivae* has not been reported to date.

IKC has been reported in wild populations of chamois (*Rupicapra* spp.), Alpine ibex (*Capra ibex*), Iberian ibex (*Capra pyrenaica*), mouflon (*Ovis gmelini*), and muskox from most European mountain ranges [7,10,19,20,27,28,29,30,31,32,33], being considered one of the main diseases of wild mountain ruminants [27,28,29,30,31,33,34]. IKC morbidity usually ranges between 50% and 90%, with mortality between 15% and 20% due to starvation or falling from cliffs, as blindness hinders the animal from finding food and moving around safely. However, mortality can reach 30% in IKC outbreaks [35,36,37], and the combination of natural mortality and disease management through removal of the affected individuals can have significant demographic effects [38]. The obvious ocular lesions caused by IKC raise social concern, as wild mountain ruminants are ecologically, culturally, and economically relevant [7]. Conversely, despite sporadic case reports and seasonal local outbreaks, IKC seems to have a lower population impact in deer species, with mostly a rather unclear aetiology [3,39,40,41,42]. Although *M. conjunctivae* has been detected and infectious keratoconjunctivitis outbreaks have been reported in domestic ruminants in North America, Asia, and Oceania [9,16,21,43,44,45,46,47,48,49,50], to the authors’ knowledge, reliable data on *M. conjunctivae* prevalence in wild ruminants is scarce outside of Europe’s main mountain ranges.

*Mycoplasma conjunctivae* interspecific transmission can happen in the meadows of mountain ecosystems, seasonally shared and maintained by the host community of domestic and wild ruminants from spring to autumn [4,19,51,52,53]. Livestock, and particularly sheep (*Ovis aries*) rather than goats (*Capra hircus*), have been traditionally considered the reservoir for *M. conjunctivae* and the main source of infection for wild ungulates, who would act as spill-over hosts in mountain ecosystems [21,27]. In this scenario, asymptomatic infections occur more frequently in domestic sheep, with wild hosts suffering more severe clinical signs [7,18,21,54,55]. However, *M. conjunctivae* has been isolated from the eyes of healthy wild mountain ungulates, and epidemiological cycles of *M. conjunctivae* in wild host populations without the participation of domestic small ruminants have been demonstrated [17,19,20,32,54]. Consequently, wild ungulates can either develop severe IKC, usually as spill-over hosts when they are exposed to new *M. conjunctivae* strains, or be asymptomatic carriers of *M. conjunctivae*, acting as reservoir and maintenance hosts of endemic *M. conjunctivae* strains, with or without the involvement of livestock [7,17,19,20,27,32,56,57].

Iberian ibex is a medium-sized mountain ruminant whose populations are currently expanding in the Iberian Peninsula, being recently reintroduced in the French Pyrenees. One of the most abundant and genetically diverse populations of Iberian ibex inhabits the Natural Space of Sierra Nevada (NSSN) in southeastern Spain [58,59,60,61,62]. Although sarcoptic mange, caused by the mite *Sarcoptes scabiei*, is probably the most widespread and relevant disease affecting Iberian ibex [33], sporadic IKC cases and outbreaks have also been reported in this species, including the NSSN population [20,29,30]. However, the role of Iberian ibex in IKC dynamics in multi-host communities in mountain habitats has been poorly studied.

This study aims at assessing *M. conjunctivae* infection dynamics in the Iberian ibex and a seasonally sympatric domestic small ruminant host community of the NSSN, to elucidate the epidemiological role of the small ruminant species in the IKC maintenance and dispersal in the area.

## 2. Materials and Methods

### 2.1. Study Area

This study was performed in the NSSN (172,318 Ha), in Andalusia (southeastern Spain). The NSSN is formed by the core National Park (85,883 Ha) and the surrounding Natural Park (86,355 Ha). Nestled in the central part of the Baetic mountain range, the altitude of the Sierra Nevada massif ranges from 300 to 3482 m above sea level at the Mulhacén, the highest summit of the Iberian Peninsula. The climate is eminently Mediterranean, although its high mountain condition also confers characteristics of a cold continental climate in the highest areas [63]. Two different study areas were defined within the NSSN: Poniente (PO) in the northwest and Alpujarra (ALP) in the southwest (Figure 1).

Iberian ibex is the most abundant ungulate species of Sierra Nevada, although other wild ungulates, such as red deer (*Cervus elaphus*) and wild boar (*Sus scrofa*), can also be found. Domestic ruminants, namely cattle (*Bos taurus*), sheep, and goats, also share the NSSN areas with wild species during the grazing period (spring to autumn), as well as domestic and small groups of feral horses all year round.

### 2.2. Sampling

Eye swabs were collected from both eyes of 147 ibexes (46 subadults, 11 females and 35 males, and 101 adults, 22 females and 79 males) captured from 2015 to 2017, either through teleanaesthesia with a combination of 3 mg/kg of xylazine and 3 mg/kg of ketamine [64] in PO (*n* = 125; 37 subadults, 6 females and 31 males, and 88 adults, 12 females and 76 males) or using drive-nets [65] in ALP (*n* = 22; 9 subadults, 5 females and 4 males, and 13 adults, 10 females and 3 males) in the NSSN (Table 1) within the regular monitoring and management plan of the population. Age was determined based on horn ring count [66], and the sampled ibexes were classified as subadults (0–3 years) or adults (older than 3 years). After capture and during handling, each ibex was identified with a plastic collar and a subcutaneous electronic chip, thus avoiding unintentionally sampling it twice. 

In 2017, both eyes of 169 adult domestic small ruminants (101 sheep, 99 females and 2 males, and 68 goats, 53 females and 15 males) from two domestic flocks sharing seasonal grazing pastures with Iberian ibexes in the NSSN were also sampled in PO (68 sheep, 66 females and 2 males) and in ALP (33 sheep, all females, and 68 goats, 53 females and 15 males). The PO flock migrates over 100 km to overwinter outside of the NSSN, whereas the ALP flock remains in the area all year round, performing only altitudinal migration. The flocks were sampled twice, first in early spring before moving to the mountain meadows shared with Iberian ibexes and then in late autumn, immediately after their return to the lower plain stalls.

The eye samples were obtained from the conjunctival sac below the nictitating membrane with sterile cotton swabs without medium (Figure 2) and frozen at −20 °C within 24 h of collection. The presence of eye lesions, sex, age, date, and location were recorded for each sampled individual.

### 2.3. Molecular Detection of M. conjunctivae DNA

The eye swabs were thawed, cut, and mixed for one minute with 0.5 mL of lysis buffer (100 mM of Tris-HCl, pH 8.5, 0.05% Tween 20, 0.24 mg/mL of proteinase K) in sterile tubes. Both swabs of each individual were introduced in the same sterile tube and analysed together. The lysates of the cells were obtained by incubating the tubes at 60 °C for 60 min, and then proteinase K was inactivated at 97 °C for 15 min. The resulting lysates were directly used as test samples for the detection of *M. conjunctivae* DNA using a previously described TaqMan real-time PCR (qPCR) with primers LPPS-TM-L, LPPS-TM-R, and probe LPPS-TM-FT [67]. Briefly, 2.5 μL of the sample lysates, 900 nM of each forward and reverse primer, 300 nM of the probe, 12.5 μL of TaqMan12x Universal PCR Master Mix (Applied Biosystems, Warrington, UK), and an exogenous internal positive control (IPC, Applied Biosystems, Warrington, UK) were introduced in each reaction well and topped with nuclease-free water to reach a total volume of 25 μL. Cycling conditions were set for 40 cycles at 95 °C for 15 s and 60 °C for 1 min, with pre-cycling steps of 50 °C for 2 min and 95 °C for 10 min. The threshold cycle (Ct) of each sample was defined as the number of cycles at which the fluorescent signal of the reaction crossed the 0.05 threshold [67].

### 2.4. Statistical Analyses 

First, *M. conjunctivae* infection in Iberian ibexes was assessed as a response-dependent variable using generalised linear models (GLM) with binomial logistic distribution, and the year of sampling (2015, 2016, and 2017), period as a function of sympatry with livestock (before: 1 January to 31 May; during: 1 June to 31 October; after: 1 November to 31 December), area (PO and ALP), sex (male and female), and age category (subadult and adult) as predictor independent variables, including all of the possible two-way interactions. A post hoc Fisher’s Test was performed to statistically assess the differences in *M. conjunctivae* infection among the categories within the variables with more than two categories (namely, year of sampling, period of sympatry, and the interactions between variables that were significant according to the GLM). Finally, a model selection procedure based on the Akaike information criterion (AIC) [68] was performed for the Iberian ibex *M. conjunctivae* prevalence models. The model with the lowest AIC was retained, and the remaining competing models were ordered according to their Akaike differences (Δi) up to Δi = 2 with respect to the best model (lowest AIC). In a second analysis, a repeated-measures ANOVA GLM, also with binomial distribution, was fitted for *M. conjunctivae* infection in livestock as a response-dependent variable, with period (before and after sharing pastures with ibexes) and area (PO and ALP) as predictor independent variables, including their additive and two-way interaction effects. The individual was included as a repeated factor. The uneven distribution of host species and sex among the areas sampled did not allow for their inclusion as factors in the GLM. Therefore, Fisher’s exact tests were performed to study the differences in *M. conjunctivae* infections among livestock according to the species (goat and sheep) and sex (female and male). All of the statistical analyses were performed with the R software 4.2.2 [69]. Statistical significance was set at *p* < 0.05 for all of the tests. 

### 2.5. Mycoplasma Conjunctivae Subtyping and Cluster Analyses

The lppS gene of *M. conjunctivae* encodes for a membrane lipoprotein involved in adhesion to the cells of the host [70], with a variable domain that can be used for *M. conjunctivae* subtyping and molecular epidemiology analyses [19,70]. For cluster analyses, samples from this study with Ct values lower than 33 at the *M. conjunctivae* PCR were considered for sequencing, according to the previously reported protocol [19]. The lppS gene sequences were obtained using a nested PCR, as previously described [19,52,54,70]. The PCR products were then purified with High Pure PCR Product Purification Kit (Roche Diagnostics, Rotkreuz, Switzerland). The sequences were determined with the sequencing primers Ser_start2, Ser_start0, and Ser_end0 (Appendix A) using the BigDye termination cycle sequencing kit (Applied Biosystems, Forster City, CA, USA). The resulting sequences were trimmed to contain the region that comprises the nucleotide positions 3935–5035 of the lppS gene from the *M. conjunctivae* type strain HRC/581 (GenBank acc. number AJ318939), which corresponds to the variable lppS domain and flanking regions. Alignment and editing of the sequences were performed with the BioEdit 7.2.5. software. A phylogenetic analysis of the sequences was then performed through the generation of cluster analysis trees built using the UPGMA statistical method and performing 1000 bootstrap replications [71]. The phylogenetic tree was built using MEGA 7.0.26 software [72], including the *M. conjunctivae* sequences obtained in this study, those previously described in a captive Iberian ibex population in the NSSN [20], and the type strains HRC/581T, My 66 95 s, MY 7 96 g, and 38 s.

## 3. Results

No clinical signs or ocular lesions compatible with IKC were observed in any of the animals sampled. Nevertheless, *M. conjunctivae* was identified in all three species sampled (Iberian ibex, domestic sheep, and domestic goat).

### 3.1. M. conjunctivae Infection in Iberian ibex

The best model explaining *M. conjunctivae* prevalence in Iberian ibex included the interaction of sampling year and age category (which was included in all five models with and Akaike weight > 0.01) and the additive effects of period of sympatry with livestock and host sex (Table 2). The factors year of sampling (z value = 3.717, *p* = 0.000201), age category (z value = 3.105, *p* = 0.001900), and their interaction (z value = −3.105, *p* = 0.001901) were significant (Table 3), while period (z value = −1.868, *p* = 0.061718) and sex (z value = −1.773, *p* = 0.076279) had only a marginal effect (Table 4).

### 3.2. M. conjunctivae Infection in Domestic Small Ruminants

The GLMs did not detect statistically significant differences in *M. conjunctivae* prevalence in livestock related to the period of sympatry with Iberian ibex or area, and the Fisher’s exact test did not find statistically significant differences between females and males. Conversely, the Fisher’s exact test found a significantly higher *M. conjunctivae* prevalence in goats than in sheep, both when including all of the samples pooled together (goat 22/136, 16.2%, CI95 10.0–22.4; sheep 13/201, 6.5%, CI95 3.1–9.9; odds ratio = 2.781989, *p* = 0.005839) and for the samples obtained before sharing summer grazing pastures with Iberian ibex (goat 11/68, 16.2%, CI95 7.4–24.9; sheep 5/100, 5.0%, CI95 0.7–9.3; odds ratio = 3.637525, *p* = 0.0221). However, the *M. conjunctivae* prevalence was not significantly different between the two domestic species after the summer grazing period (goat 11/68, 16.2%, CI95 7.4–24.9; sheep 8/101, 7.9%, CI95 2.7–13.2; odds ratio = 2.232306, *p* = 0.1351; Table 5). 

### 3.3. Clustering of M. conjunctivae Strains

The dendrogram tree identified five different *M. conjunctivae* strain clusters, with one belonging to the previously reported strains identified in Iberian ibexes from the NSSN [20] and four clusters from the samples analysed in this study (Figure 3). Cluster A included samples of Iberian ibexes from both study areas (2015 and 2017) and domestic goats from ALP; cluster B was formed by samples from Iberian ibexes from PO (2015 and 2017) and domestic goats from ALP; cluster C had samples of Iberian ibexes from both study areas (2015 and 2016) and domestic sheep from PO; and, finally, cluster D was composed of strains identified from samples of domestic sheep from PO only. None of the positive samples of sheep from ALP could be sequenced due to the low presence of DNA (Figure 3). 

## 4. Discussion

The consistent detection throughout the three-year study period of up to four genetic clusters of *M. conjunctivae* in Iberian ibex and domestic sheep and goats demonstrated the circulation and maintenance of this pathogen at the wildlife–livestock interface of the NSSN. However, all of the positive individuals were asymptomatic carriers without clinical signs or ocular lesions compatible with IKC. Although asymptomatic carriers can occur both in epizootic and endemic situations [20,54], the regular detection of *M. conjunctivae* during the study period combined with the consistent presence of asymptomatic carriers indicates that *M. conjunctivae* infection is not only maintained but endemic in the small ruminant host community of the NSSN [18,19,20]. While this was the endemic situation in 2015–2017, the lack of a continued specific IKC monitoring program only allows us to assume that endemicity has been maintained since then.

The detection of three of the four *M. conjunctivae* clusters identified both in wild and domestic hosts (Figure 3) suggests that they are equally relevant in the maintenance and circulation of this pathogen. However, the detection in 2017 in a female yearling of the same *M. conjunctivae* strain from cluster B detected in the same area (PO) in 2015 in a two-year-old male (Figure 3) indicated that this strain circulated in the Iberian ibex population from the area for at least three years, agreeing with previous reports on the maintenance of *M. conjunctivae* in wild hosts without the involvement of livestock [17,19,20,32,54]. *Mycoplasma conjunctivae* endemicity in the NSSN Iberian ibex population seemed to be maintained mainly thanks to its circulation in naïve subadults getting infected, as shown by the higher subadult prevalence compared to adults in 2015 and 2016 (Table 3). Furthermore, young individuals can contribute to the spread of *M. conjunctivae* and other pathogens during their dispersion movements after leaving the familiar group where they were born [73]. The role of younger individuals in the maintenance of *M. conjunctivae* has already been reported in endemic situations, both in wild and domestic hosts [19,20,54]. In such endemic scenarios, epizootic outbreaks occur when the infection spreads and affects adults [7,17,19,27,28,32,54,74], as occurred in the NSSN in 2017, with a significant prevalence increase in adults leading to a significantly higher overall prevalence in Iberian ibex (Table 3) without apparent morbidity [20,75]. Whether these outbreaks occur through contact with new strains, a decrease in herd immunity, or other host-, pathogen-, or environment-related factors remains to be ascertained.

On the other hand, the detection of cluster D strains only in domestic sheep from PO, both before and after the summer grazing period (Figure 3), confirmed the repeatedly reported *M. conjunctivae* maintenance and reservoir host role of livestock [7,18,19,21,76]. Although sheep has been considered a more relevant reservoir host than goats for *M. conjunctivae* [18,19,21,76], in the NSSN, the prevalence was higher in goats than in sheep (Table 5), revealing that goats are probably more and at least as relevant as sheep in the circulation, spread, and maintenance of *M. conjunctivae* in the study area. Therefore, the drivers for a species or population to be able to act as a reservoir for *M. conjunctivae* do not seem to be related to intrinsic species–specific features but rather the host community’s composition and the interaction among the host species and population, the pathogen, and the environment [44,77,78,79]. Because the PO flock overwinters outside of the NSSN, this strain could probably have been acquired outside of the NSSN and introduced in the system during the seasonal migration. While this newly acquired strain had not yet spread to the Iberian ibex population or other livestock flocks, its maintenance in the migrating flock makes it a matter of time until the transmission of this new strain to other susceptible hosts in the NSSN. Such a threat could modify the low pathogenic endemic status of IKC identified in 2015–2017, as the introduction of new strains could not only increase the diversity of *M. conjunctivae* in the NSSN but also produce more intense clinical signs and pathogenicity. Establishing integrated wildlife monitoring in the NSSN, encompassing both wildlife population monitoring and health surveillance [79], including IKC, would therefore be essential to assess the potential introduction of new *M. conjunctivae* strains and/or changes in the epidemiological status of IKC.

Although both Iberian ibex and livestock in the NSSN could thus maintain independently *M. conjunctivae* without the involvement of other host species, most of the clusters were identified in both wild and domestic hosts, suggesting interspecific transmission and shared epidemiological cycles at the wildlife–livestock interface, as described in other host communities [17,18,19,52,54,80]. Moreover, the strains from clusters A, B, and C were each identified in just one of the study areas in livestock, while in Iberian ibex, they were either identified in both study areas (clusters A and C) or in the area where it had not been detected in livestock (cluster B; Figure 3). The cross-transmission between Iberian ibexes and domestic small ruminants probably occurs during the period of sympatry in the summer grazing pastures [19,51,52,53,81], favoured by the increase and concentration of susceptible hosts [82] and the seasonal presence of eye-frequenting flies that can transmit *M. conjunctivae* [23,24]. In the NSSN, this explanation is further supported by the lower prevalence of *M. conjunctivae* found in the Iberian ibex population before the summer grazing period, with the summer prevalence increase lasting throughout the fall (Table 4).

The results suggest not only that interspecific transmission of *M. conjunctivae* occurs in the NSSN, but also that the Iberian ibex population, widely and densely distributed in the area [61,62], maintains the infection and is a nexus for epidemiological interaction among flocks of domestic small ruminants that are otherwise spatially segregated in the mountain massif, occupying separately distant pasture areas that are difficult to connect by livestock but not by the ibex population. This finding is relevant not only for the epidemiology of *M. conjunctivae*, but also because it opens new approaches to the dynamics of other shared pathogens at the wildlife–livestock interface, including notifiable diseases. This should be considered for integrated monitoring at the wildlife–livestock interface in the NSSN, including not only health surveillance but also population monitoring [79,83].

## 5. Conclusions

To summarise, *M. conjunctivae* infection was asymptomatic and endemic in the small ruminant host community of the NSSN, involving Iberian ibex, domestic goat, and domestic sheep. All of the host species could maintain independently the circulation of this pathogen, although most strains were shared during the summer grazing season by wildlife and livestock in sympatry. In Iberian ibex, endemicity was mainly maintained by naïve subadults, with epizootic outbreaks occurring when the infection spread to adults in 2017. Among livestock, goat was at least as important as sheep in the maintenance of *M. conjunctivae*. Iberian ibex should be considered a key species in the maintenance and spread of infection in shared high mountain areas, not only because it is capable of maintaining the infection intraspecifically, but also because of its probable role in the IKC spread between spatially segregated livestock flocks grazing in separate areas of the NSSN. Implementing integrated wildlife monitoring is essential to detect any changes in the status of IKC in the NSSN. These results should be useful for both researchers and population and health managers of domestic and wild ruminants in mountain ecosystems, as they demonstrate the need to address the management of IKC and other shared pathogens considering the multi-host scenario of epidemiological interactions.

## Figures and Tables

**Figure 1 vetsci-11-00217-f001:**
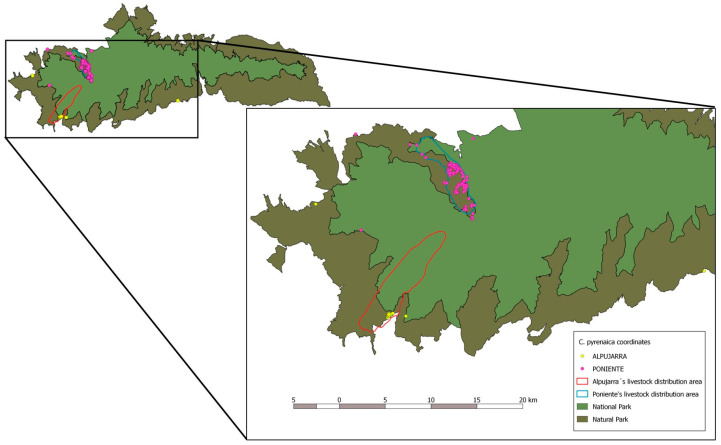
Study area in the Natural Space of Sierra Nevada, including the distribution of the sampled animals and the spatial relation between the wild and domestic sampled animals.

**Figure 2 vetsci-11-00217-f002:**
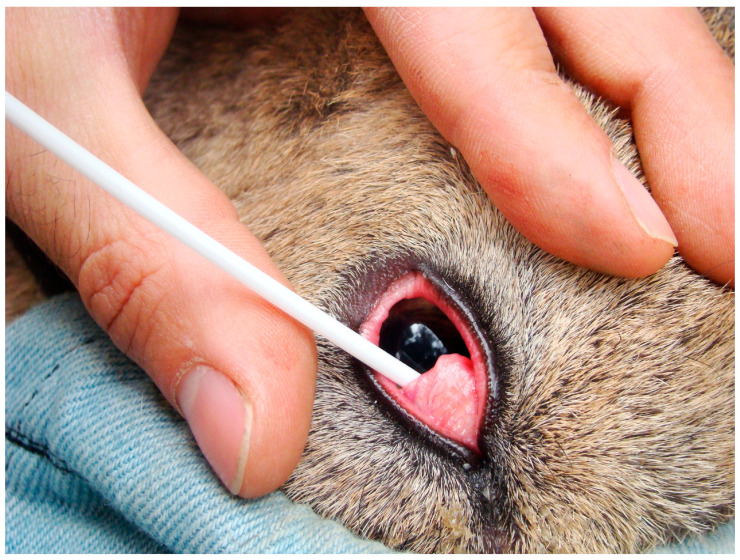
Swab sampling for *M. conjunctivae* detection from the conjunctival sac below the nictitating membrane in an Iberian ibex from the Natural Space of Sierra Nevada.

**Figure 3 vetsci-11-00217-f003:**
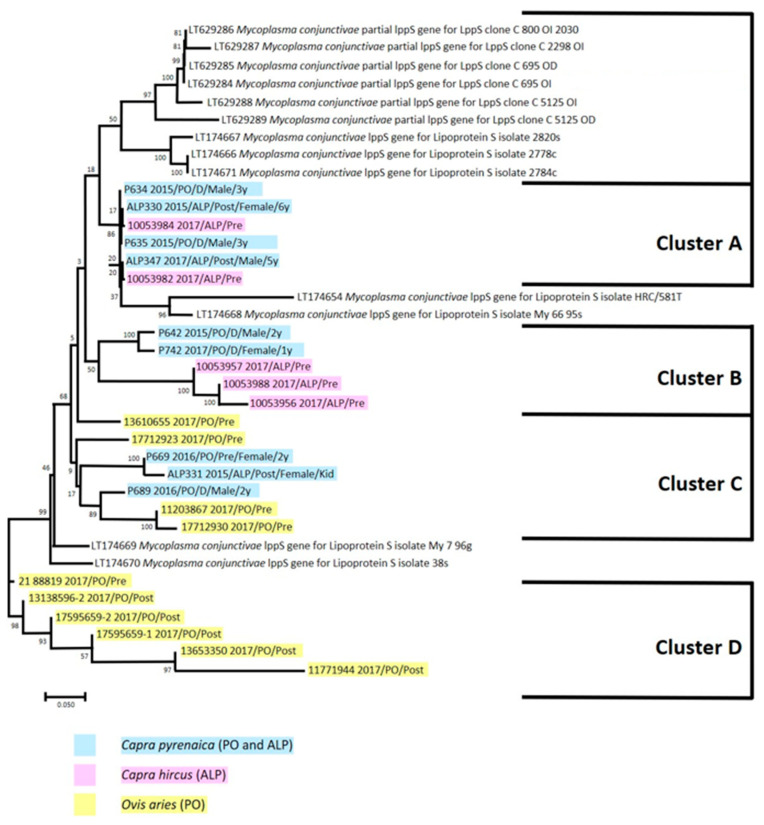
Cluster analysis tree of the *M. conjunctivae* strains detected in the Iberian ibexes and domestic livestock from the Natural Space of Sierra Nevada, including those previously reported in the area [20].

**Table 1 vetsci-11-00217-t001:** Wild and domestic small ruminants sampled for this study in the Natural Space of Sierra Nevada (NSSN). ALP = Alpujarra; PO = Poniente; F = female; M = male.

	2015				2016				2017				2015–2017								
	ALP		PO		ALP		PO		ALP		PO		ALP			PO			NSSN		
	F	M	F	M	F	M	F	M	F	M	F	M	F	M	Total ALP	F	M	Total PO	F	M	Total
**Iberian ibex**																					
Subadults	3	2	1	11	1	2	2	18	1	0	3	2	5	4	9	6	31	37	11	35	46
Adults	5	3	12	14	2	0	0	30	3	0	0	32	10	3	13	12	76	88	22	79	101
**Subtotal ibex**	**8**	**5**	**13**	**25**	**3**	**2**	**2**	**48**	**4**	**0**	**3**	**34**	**15**	**7**	**22**	**18**	**107**	**125**	**33**	**114**	**147**
**Domestic sheep**									33	0	66	2	33	0	33	66	2	68	99	2	101
**Domestic goat**									53	15	0	0	53	15	68	0	0	0	53	15	68
**Subtotal livestock**									**86**	**15**	**66**	**2**	**86**	**15**	**101**	**66**	**2**	**68**	**152**	**17**	**169**

**Table 2 vetsci-11-00217-t002:** Model selection to explore *M. conjunctivae* in the Iberian ibexes sampled in the NSSN. Only the models with an Akaike weight (ωi) 0.01 are shown.

Model	K	AIC	Δi	ωi
**Year x age class + period + sex**	**7**	**116.60**	**0.00**	**0.40**
Year x age class + area + sex	6	118.04	1.44	0.20
Year x age class + period + area + sex	8	118.66	2.06	0.14
Year x age class	4	118.79	2.19	0.14
Year x age class + period + area	7	119.67	3.07	0.09

K = number of parameters, Δi = difference in AIC with respect to the best model, ωi = Akaike weight. The best model is indicated in bold.

**Table 3 vetsci-11-00217-t003:** Detection of *M. conjunctivae* DNA through PCR in the Iberian ibexes from the Natural Space of Sierra Nevada (NSSN), according to year and age class.

		2015	2016	2017	2015–2017
**Subadults**	Positive/sampled	5/17	3/23	2/6	10/46
	Prevalence	29.4% ^a^	13.0% ^a^	33.3%	21.7%
	Confidence interval 95%	7.8–51.1	0.0–26.8	0.0–71.1	9.8–33.7
**Adults**	Positive/sampled	1/34	0/32	15/35	16/101
	Prevalence	2.9% ^bx^	0.0% ^bx^	42.9% ^y^	15.8%
	Confidence interval 95%	0.0–8.6	0.0–0.0	26.5–59.3	8.7–23.0
**Total**	Positive/sampled	6/51	3/55	17/41	26/147
	Prevalence	11.8% ^x^	5.5% ^x^	41.5% ^y^	17.7%
	Confidence interval 95%	2.9–20.6	0.0–11.5	26.4–56.5	11.5–23.9

^a,b^: the prevalences with different superscripts are significantly (*p* < 0.05) different from each other between age classes; ^x,y^: the prevalences with different superscripts are significantly (*p* < 0.05) different from each other between years.

**Table 4 vetsci-11-00217-t004:** Detection of *M. conjunctivae* DNA through PCR in the Iberian ibexes from the Natural Space of Sierra Nevada (NSSN), according to period and sex. Pre = before sharing grazing pastures with domestic livestock (01/01 to 31/05); During = while sharing grazing pastures with domestic livestock (01/06 to 31/10); Post = after sharing grazing pastures with domestic livestock (01/11 to 31/12).

PERIOD	Pre	During	Post
**Positive/sampled**	6/46	15/74	5/27
**Prevalence**	13.0% ^a^	20.3% ^b^	18.5% ^b^
**Confidence interval 95%**	3.3–22.8	11.1–29.4	3.9–33.2
**SEX**	**Female**	**Male**	
**Positive/sampled**	5/33	20/114	
**Prevalence**	15.2%	17.5%	
**Confidence interval 95%**	2.9–27.4	10.6–24.5	

^a,b^: the prevalence previous to sharing grazing pastures with domestic livestock in summer was marginally different (*p* = 0.061718) to the prevalence during and after sharing grazing pastures with livestock.

**Table 5 vetsci-11-00217-t005:** Detection of *M. conjunctivae* DNA through PCR in the sheep and goats from the Natural Space of Sierra Nevada (NSSN), according to the species, period, study area, and sex. Pre = before sharing seasonal grazing pastures with Iberian ibexes; Post = after sharing seasonal grazing pastures with Iberian ibexes; ALP = Alpujarra; PO = Poniente.

	Pre							Post						
	ALP			PO			Total Pre	ALP			PO			Total Post
	Female	Male	Subtotal ALP	Female	Male	Subtotal PO		Female	Male	Subtotal ALP	Female	Male	Subtotal PO	
**Goat**	10/53	1/15	11/68	0/0	0/0	0/0	11/68 ^a^	11/53	0/15	11/68	0/0	0/0	0/0	11/68
	18.9%	6.7%	16.2%				16.2%	20.8%	0.0%	16.2%				16.2%
	8.3–29.4	0.0–19.3	7.4–24.9				7.4–24.9	9.8–31.7	0.0–0.0	7.4–24.9				7.4–24.9
**Sheep**	2/33	0/0	2/33	3/65	0/2	3/67	5/100 ^b^	4/33	0/0	4/33	4/66	0/2	4/68	8/101
	6.1%		6.1%	4.6%	0.0%	4.5%	5.0%	12.1%		12.1%	6.1%	0.0%	5.9%	7.9%
	0.0–14.2		0.0–14.2	0.0–9.7	0.0–0.0	0.0–9.4	0.7–9.3	1.0–23.3		1.0–23.3	0.3–11.8	0.0–0.0	0.3–11.5	2.7–13.2
**TOTAL**	12/86	1/15	13/101	3/65	0/2	3/67	16/168	15/86	0/15	15/101	4/66	0/2	4/68	19/169
	14.0%	6.7%	12.9%	4.6%	0.0%	4.5%	9.5%	17.4%	0.0%	14.9%	6.1%	0.0%	5.9%	11.2%
	6.6–21.3	0.019.3	6.3–19.4	0.0–9.7	0.0–0.0	0.0–9.4	5.1–14.0	9.4–25.5	0.0–0.0	7.9–21.8	0.3–11.8	0.0–0.0	0.3–11.5	6.5–16.0

^a,b^: the prevalences with different superscripts are significantly (*p* < 0.05) different from each other between species.

## Data Availability

All relevant data are published in this article.

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
