# Peer review of "Wildlife–Livestock Host Community Maintains Simultaneous Epidemiologic Cycles of Mycoplasma conjunctivae in a Mountain Ecosystem"

_vetsci, 2024, doi:10.3390/vetsci11050217_

Round 1
Reviewer 1 Report
Comments and Suggestions for Authors
Dear Authors,
The manuscript titled "Wildlife-livestock host community maintains simultaneous epidemiologic cycles of Mycoplasma conjunctivae in a mountain ecosystem" is an interesting study on the role of Mycoplasma conjunctivae in keratoconjunctivitis among small ruminants (wild and domestic caprines). You have done a cluster and subcluster analysis on Mycoplasma conjunctivae strains using LppS that encodes a lipoprotein that contributes to the adhesion of the bacterium to the host cells. The variable domains within LppS could be used for M. conjunctivae subtyping and molecular epidemiology analyses. Please find my comments as follows:
Introduction:
I suggest comparing the prevalence of keratoconjunctivitis caused by Mycoplasma conjunctivae (in ruminants) in your area with other parts of the world!
- You should describe how important it is to detect Mycoplasma conjunctivae earlier in the contaminated animals! Also, please explain if Mycoplasma conjunctivae is resistant to treatments such as antibiotics! If yes, why?
- Line 54: Please discuss, in a few lines, if the disease can be transmitted to humans or not.
Line 180: adhesion of what? please explain more!
- Line 183: Did you have any reference positive control? How did you choose this Ct (33) as the cutoff for positivity? Have you tested the understudied animals with any immunological assays such as ELISA, etc?
Reviewer 2 Report
Comments and Suggestions for Authors
The article describes the problem well and documents it with a sufficient bibliography. However, I ask the authors whether the data collected between 2015 and 2017 are currently representative of the epidemiological situation of Natural Space of Sierra Nevada. I suggest to improve in introduction or in discussion, it is possible, some recent data about the disease situation.
Reviewer 3 Report
Comments and Suggestions for Authors
Summary: The introduction details the etiopathogenesis and clinical significance of infectious keratoconjunctivitis (IKC) by the agent Mycoplasma conjunctivae with particular focus on the Iberian ibex. Over the course of three years, the authors gathered samples via eye swabs of 147 wild ibex captured during the course of regular monitoring and management of the population as well as an additional 169 domestic ruminants (sheep and goats). Swabs were obtained from the conjunctival sac of both eyes from each animal. RT-PCR was used for detection and subtyping of Mycoplasma conjunctivae nucleic acids. Statistical analysis was performed using generalized linear models, repeated measures ANOVA, and Fisher’s exact t-test. M. conjunctivae was found in all three species although clinical signs were not seen in any individual, suggesting maintenance of endemic infection in the species in this region. Genetic clusters of M. conjunctivae were analyzed and compared between the different populations as well as different sexes and age groups within the populations. In ibex, M. conjunctivae strains were mostly maintained by younger animals. Strains were more likely to be spread and exchanged between populations during summer grazing periods.
Review: The information included in the introduction is sufficient for understanding the importance of the disease and informing the reader of necessary details regarding the target animal population. The sample size is sufficient for this study. The statistical analyses are very thorough. The epidemiological analyses are intriguing and show the distribution and spread of different strains of M. conjunctivae within and between species. The data supports the authors’ conclusions that M. conjunctivae is maintained endemically by wild ibex as well as domestic sheep and goats.
Overall, the study design, methods, statistics, and conclusions are all sufficient. The paper is ready for publication pending some minor corrections related to figure formatting and grammar.
Is there a chance that any of the individual ibex studied in 2015 were also captured and studied in 2016 or 2017? This seems unlikely given that the swabs were collected alongside the regular monitoring and management program, but a brief sentence should be included in Section 2.2 of the Materials and Methods section to clarify whether the ibex populations were examined only once or if there is a chance that some of the individuals may have been tested repeatedly.
In general, the formatting for the tables should be reviewed prior to publication with special care to keep each table on the same page (e.g., Table 1 extends from page 3 to page 4, and Table 4 expands from page 7 to page 8). Table 4 in particular would benefit from being displayed in landscape orientation on a separate page. The image quality for Figure 3 is poor, and the text is blurry and difficult to read.
Minor grammatical corrections:
Line 144: “real time PCR” should be “real-time PCR”
Line 200-201: “No clinical signs nor ocular lesions compatible with IKC were observed in none of 200 the animals sampled” should be “No clinical signs or ocular lesions compatible with IKC were observed in any of the 200 the animals sampled”
